# RAPSODI: Radiosonde Atmospheric Profiles from Ship and island platforms during ORCESTRA, collected to Decipher the ITCZ

Marius Winkler¹, Marius Rixen¹, Florent Beucher², Fleur Couvreux², Chaehyeon C. Nam³, Philippe Peyrillé², Hauke Schmidt¹, Hans Segura¹, Karl-Hermann Wieners¹, Ezri Alkilani-Brown⁴, Abdou Aziz Coly⁵, Giovanni Biagioli⁶, Michael M. Bell⁵, Ester Brito⁶, Emma Chauvin², Julie Capo², Delián Colón-Burgos⁻, Akeem Dawes⁶, Jose Carlos da Luz⁶, Zekican Demiralay¹⁶, Vincent Douet¹⁰, Vincent Ducastin², Clarisse Dufaux⁶, Jean-Louis Dufresne⁶, Florence Favot², Thomas Fiolleau¹¹, Emilie Fons⁶, Geet George¹², Helene M. Gloeckner¹, Suelly Gonçalves¹², Laurent Gouttesoulard¹¹, Lennéa Hayo¹, Wei-Ting Hsiao³, Sarah Kennison³, Michael Kopelman³, Tsung-Yung Lee³, Enora Le Gall⁶, Mateo Lovato¹³, Emily Luschen¹⁴, Nicolas Maury⁶, Brett McKim⁶, Louis Netz¹¹, Diouf Ousseynou⁵, Karsten Peters-von Gehlen¹², Chavez Pope⁰, Basile Poujol⁶, Niwde Rivera Maldonado¹⁵, Nina Robbins-Blanch¹⁰, Nicolas Rochetin⁶, Daniel Rowe⁰, Paula Romero Jure⁴, James H. Ruppert Jr. ¹⁴, Jairo Segura Bermudez¹, Jarrett C. Starr³, Martin Stelzner¹, Connor Stoll³, Macintyre Syrett¹, Abraham Tekoe³, Jeremie Trules¹, Colin Welty¹, Daniel Klocke¹, Raphaela Vogel¹, Sandrine Bony⁶, Allison A. Wing³, and Bjorn Stevens¹

**Correspondence:** Marius Winkler (marius.winkler@mpimet.mpg.de)

<sup>&</sup>lt;sup>1</sup>Max-Planck-Institut für Meteorologie, Hamburg, Germany

<sup>&</sup>lt;sup>2</sup>Centre National de Recherches Météorologiques, University of Toulouse, Météo-France, CNRS, Toulouse, France

<sup>&</sup>lt;sup>3</sup>Department of Earth, Ocean and Atmospheric Science, Florida State University, Tallahassee, FL, USA

<sup>&</sup>lt;sup>4</sup>School of Earth and Environment, University of Leeds, Leeds, United Kingdom

<sup>&</sup>lt;sup>5</sup>Agence Nationale de l'Aviation Civile et de la Météorologie, Dakar, Sénégal

<sup>&</sup>lt;sup>6</sup>Laboratoire de Météorologie Dynamique/IPSL, Sorbonne Université, École Normale Supérieure, École Polytechnique, CNRS, Paris, France

<sup>&</sup>lt;sup>7</sup>Department of Atmospheric Science, Colorado State University, Fort Collins, CO, USA

<sup>&</sup>lt;sup>8</sup>Instituto Nacional de Meteorologia e Geofísica, Sal, Cape Verde

<sup>&</sup>lt;sup>9</sup>Caribbean Institute for Meteorology and Hydrology, Barbados

<sup>&</sup>lt;sup>10</sup>Institute Pierre-Simon Laplace, CNRS, Paris, France

<sup>&</sup>lt;sup>11</sup>Laboratoire d'Etudes en Geophysique et Océanographie Spatiale, University of Toulouse, IRD, CNRS, CNES, UPS, Toulouse, France

<sup>&</sup>lt;sup>12</sup>Department of Geoscience and Remote Sensing, Delft University of Technology, Delft, Netherlands

<sup>&</sup>lt;sup>13</sup>Department of Electrical and Computer Engineering, Colorado State University, Fort Collins, CO, USA

<sup>&</sup>lt;sup>14</sup>School of Meteorology, University of Oklahoma, Norman, OK, USA

<sup>&</sup>lt;sup>15</sup>University of Puerto Rico, Mayagüez, Puerto Rico

<sup>&</sup>lt;sup>16</sup>Meteorological Institute, Ludwig Maximilian University, Munich, Germany

<sup>&</sup>lt;sup>17</sup>Deutscher Wetterdienst, Offenbach, Germany

<sup>&</sup>lt;sup>18</sup>Deutsches Klimarechenzentrum GmbH, Hamburg, Germany

<sup>&</sup>lt;sup>19</sup>Universität Hamburg, Hamburg, Germany

These authors contributed equally to this work

Abstract. The RAPSODI (Radiosonde Atmospheric Profiles from Ship and island platforms during ORCESTRA, collected to Decipher the ITCZ) radiosonde dataset was collected during the ORCESTRA field campaign. It is designed to investigate the mechanisms linking mesoscale tropical convection to tropical waves and to air—sea heat and moisture exchanges that regulate convection and tropical cyclone formation. The campaign began at the Instituto Nacional de Meteorologia e Geofisica (INMG) on Sal on the Cape Verde Islands, continued with ship-based observations aboard the R/V Meteor across the Atlantic, and concluded at the Barbados Cloud Observatory (BCO) in the eastern Caribbean. During the campaign, a total of 624 radiosondes were launched, capturing high-resolution profiles of temperature, humidity, pressure, and winds. This radiosonde dataset, encompassing raw, quality-controlled, and vertically gridded data, is detailed in this paper and offers a valuable resource for investigating the atmospheric structure and processes shaping tropical convection and the intertropical convergence zone (ITCZ). The complete dataset is openly available at ipfs://bafybeid7cnw62zmzfgxcvc6q6fa267a7ivk2wcchbmkoyk4kdi5z2yj2w4.

#### 1 Introduction

field study aimed at advancing the understanding of tropical meteorology and atmospheric processes¹ (Stevens et al., 2026). It brought together several subcampaigns, including for example PERCUSION² and MAESTRO³, which targeted convective organization with aircraft observations, as well as BOWTIE⁴ and PICCOLO⁵, which focused on ocean–atmosphere interactions and convective organization from the R/V Meteor, and SCORE⁶, which investigated rain evaporation at the Barbados Cloud Observatory. Through a collaborative effort across multiple subcampaigns and platforms, radiosondes were launched across the Atlantic from Sal in the east and Barbados in the west, complemented by ship-borne measurements aboard the R/V Meteor. Radiosondes, which are a central tool for observing the vertical structure of the tropical atmosphere, were included in each of the subcampaigns mentioned above, as all contributed to their funding. They had long been an essential tool for profiling the atmosphere (Lettau, 1950), but the first large-scale coordinated use in an international campaign came with the GARP Atlantic Tropical Experiment (GATE) in 1974 (Mason, 1975), after which they became indispensable for characterizing atmospheric profiles in field studies. When launched frequently across multiple platforms, radiosondes enable analyses of thermodynamic profiles, wind patterns, and moisture convergence on meso- and synoptic scales. Such networks have supported progress in understanding cloud–circulation coupling, as demonstrated by EUREC⁴A (Stephan et al., 2021), where coordinated radiosonde launches captured the rapid deepening and moistening of the boundary layer during the passage of a characteristic "fish" cloud

pattern (their Fig. 10). This episode marked a transition from shallow to deeper convection, accompanied by a pronounced reorganization of humidity and pressure structures which is an example of how sustained radiosonde observations can reveal the evolving coupling between mesoscale convection and the large-scale environment. The ORCESTRA campaign builds on

The ORganized Convection and EarthCare Studies over the TRopical Atlantic (ORCESTRA) campaign is an international

<sup>&</sup>lt;sup>1</sup>https://orcestra-campaign.org/orcestra.html

 $<sup>^2</sup>$ PERCUSION  $\equiv$  Persistent EarthCare underflight studies of the ITCZ and organized convection

<sup>&</sup>lt;sup>3</sup>MAESTRO ≡ Mesoscale organisation of tropical convection

<sup>&</sup>lt;sup>4</sup>BOWTIE ≡ Beobachtung von Ozean und Wolken – Das Trans ITCZ Experiment

<sup>&</sup>lt;sup>5</sup>PICCOLO = Process Investigation of Clouds and Convective Organization over the atLantic Ocean

<sup>&</sup>lt;sup>6</sup>SCORE ≡ Sub-Cloud Observations of Rain Evaporation

© Author(s) 2025. CC BY 4.0 License.


this approach, extending sounding observations across the tropical Atlantic, including both within and at the edge of the ITCZ, and providing continuous time-height measurements during day and night.

Over the 52-day campaign, radiosondes were launched at high temporal frequency from three platforms: the Instituto Nacional de Meteorologia e Geofísica (INMG) on Sal Island, Cape Verde, the R/V Meteor during its Atlantic transect, and the Barbados Cloud Observatory (BCO) in the western Atlantic. These profiles complement measurements from other platforms, including cloud observations from remote sensing instruments on the R/V Meteor and dropsondes released from the HALO aircraft (Gloeckner et al., 2026) by offering higher maximum height, the possibility of obtaining both ascent and descent profiles from a single launch, and continuous 3-hourly coverage including at night. In the central Atlantic, where dropsonde coverage is absent, the many R/V Meteor profiles provide unique sampling. Together, these complementary datasets enable multi-scale studies of convection, tropical wave activity, and air—sea interaction.

The radiosondes from the three platforms (BCO, INMG and R/V Meteor) are aggregated into a unified dataset called RAP-SODI. The different platforms are described in Section 2. Section 3 outlines the data acquisition and the data processing methods, while Section 4 discusses the observed atmospheric conditions. Finally, the paper presents the code and data availability in Section 5 before concluding with a summary.

#### 2 The ORCESTRA Sounding Network

This section describes the ORCESTRA radiosonde network, its sampling strategy, and its measurement setups. Over the course of the campaign (Stevens et al., 2026), a total of 624 radiosondes were launched from three platforms spanning the tropical Atlantic. The stationary sites at INMG and BCO provided repeated profiles at fixed locations, while the R/V Meteor contributed time-dependent profiles along its transect from the eastern to the western Atlantic.

In the following, the three launch platforms and their respective measurement setups are described in more detail. To illustrate the diversity of observing strategies within ORCESTRA, the radiosonde launches are grouped by their platform and associated subcampaigns:

- 1. as part of MAESTRO (*Mesoscale organisation of tropical convection*) from the INMG at Aeroporto Amílcar Cabral, Sal Island, Republic of Cabo Verde,
- 2. as part of BOWTIE (*Beobachtung von Ozean und Wolken Das Trans ITCZ Experiment*) and PICCOLO (*Process Investigation of Clouds and Convective Organization over the atLantic Ocean*) from the Research Vessel Meteor (Klocke et al., 2026) which crossed the Atlantic from east to west, and
- 3. in coordination with PERCUSION (*Persistent EarthCare underflight studies of the ITCZ and organized convection*), SCORE (*Sub-Cloud Observations of Rain Evaporation*) and PICCOLO from the Barbados Cloud Observatory at Deebles Point, St. Philip, Barbados (Stevens et al., 2016).
- Figure 1 illustrates the radiosonde launch platforms and trajectories. The ship campaign began in the eastern Atlantic at the port of Mindelo on São Vicente Island, Cape Verde, and concluded in the western Atlantic at the port of Bridgetown, Barbados.

**Figure 1.** Radiosonde trajectories across the tropical North Atlantic, with both ascending and descending segments shown. Each dot marks a location where the radiosonde passed during its flight, color-coded by height. Panels (a) and (b) show profiles launched from land-based platforms at the Barbados Cloud Observatory (BCO, Deebles Point, Barbados) and Sal Island (INMG, Cape Verde), respectively. Panel (c) shows all soundings, including those from the R/V Meteor during its meridional trajectory across the Atlantic.

Radiosondes were launched between August 9 and September 29 from the three different platforms as illustrated in Figure 2. Of the ascending profiles, 91% were successfully tracked during descent, and 48% descended below 980 hPa. However, 9% of the radiosondes lacked a measured descending branch due to early contact loss. This issue was especially evident during the early phase of the ORCESTRA campaign at INMG on Sal Island, where overlapping transmission frequencies led to radiosondes losing contact earlier than expected. Table 1 provides additional information on ascent and descent statistics, start and end dates, radiosonde vendors, and frequency settings.

Figure 2 shows that radiosonde operations were sustained throughout the ORCESTRA campaign, with activity varying between platforms. The R/V Meteor maintained the highest launch frequency, while BCO and INMG operated at lower but also consistent rates.

Table 1. Launch platform statistics and characteristics.

|                       | INMG                          | R/V Meteor                   | BCO                          |  |
|-----------------------|-------------------------------|------------------------------|------------------------------|--|
| Number of ascents     | 156                           | 327                          | 141                          |  |
| Number of descents    | 124                           | 313                          | 135                          |  |
| Start date            | 2024-08-09                    | 2024-08-14                   | 2024-09-07                   |  |
| End date              | 2024-09-11                    | 2024-09-24                   | 2024-09-29                   |  |
| Platform altitude / m | 58                            | 5.5                          | 17                           |  |
| Frequency / MHz       | 401.2, 403, 403.6, 404, 404.4 | 403, 405                     | 400.5–401.5                  |  |
| Balloon type          | Hwoyee 200g                   | TOTEX TA200-No.088 parachute | TOTEX TA200-No.088 parachute |  |
| Radiosonde vendor     | Meteomodem                    | Vaisala                      | Vaisala                      |  |

#### 2.1 INMG



A total of 156 radiosondes were launched at the INMG platform, located at  $16.73^{\circ}$  N,  $22.94^{\circ}$  W, 124 of which successfully recorded descents. The radiosondes were prepared indoor, where the air temperature was maintained at approximately  $25^{\circ}$ C. Initialization and calibration were performed using surface weather conditions measured by the INMG instruments<sup>7</sup>.

The balloons were inflated in a nearby hangar where helium was stored. The setup (Figure 3 a) included a net-covered frame to protect the balloon skin and a water bottle used as a weight for accurate filling of the balloon. Radiosondes (Meteomodem, M20) were attached with an unwinder and rope before launch (Figure 3 b). Despite proximity to an airport, no coordination with air traffic control was required.

During the first five days, many radiosondes were lost below 10 km due to interference from a nearby antenna operating on the same frequencies. Consequently, from August 9 to August 13, radiosonde launches were conducted only four times a day until a solution was implemented. After the interfering antenna was deactivated, subsequent radiosonde ascents and descents proceeded without issue. However, the limited supply of helium restricted the number of launches, averaging to four per day on non-flight days and six per day on flight days, when either the German research aircraft HALO<sup>8</sup> or the French research aircraft ATR<sup>9</sup> were conducting measurement flights as part of the PERCUSION or MAESTRO campaign.

Figure 1(b) shows the trajectories of radiosondes launched from INMG, which predominantly drift westwards under the influence of the easterly trade winds, reaching heights of up to  $27.4\,\mathrm{km}$ . On average, ascending radiosondes remained airborne for about  $110\,\mathrm{min}$  and travelled a median distance of  $45.7\,\mathrm{km}$ , while the descending radiosondes lasted around  $28\,\mathrm{min}$  with a median drift of  $12.4\,\mathrm{km}$ .

<sup>&</sup>lt;sup>7</sup>https://oscar.wmo.int/surface/index.html#/search/station/stationReportDetails/0-20000-0-08594

<sup>8</sup>https://halo-research.de

<sup>9</sup>https://www.safire.fr

**Figure 2.** Daily number of ascending (lower left triangles) and descending (upper right triangles) soundings during the 52-day ORCESTRA campaign. The color scale indicates the number of radiosondes per day operated at each platform. Instances where more descents than ascents are recorded (e.g., the second-to-last day at BCO) occur when a radiosonde launched before midnight UTC descends after midnight, shifting the descent count to the next day. The R/V Meteor's dual receiver setup allowed for a higher launch frequency.

Figure 3. Launching setups on each platform during the ORCESTRA campaign. (a) INMG platform on Sal Island, Cape Verde: balloon filling with a water bottle weight and protection from a net-covered frame. (b) Radiosonde launch in progress at INMG. (c) R/V Meteor automatic launcher with ceiling-hung ball for target balloon size. (d) Launcher opening for balloon release from the port side. (e) BCO launcher with green tarpaulin frame for balloon sizing and skin protection. (f) Launch position at BCO near the cliff edge. Credits: (a,b) Tristan Vostry; (c,d) Allison Wing; (e) Nina Robbins-Blanch; (f) Marius Winkler.

# 2.2 R/V Meteor

The Research Vessel (R/V) Meteor departed from the port of Mindelo on São Vicente Island, Cape Verde, on August 16 and arrived in Bridgetown, Barbados, on September 24 after spending 40 days at sea. Six radiosondes were launched while still in harbor prior to departure on August 16. In addition to generally steaming westward, the Meteor also performed north/south transects in the eastern, central, and western parts of the tropical Atlantic. During the campaign, a total of 327 radiosondes were launched from the R/V Meteor, with 318 successful descents recorded. The radiosondes were prepared and placed in an automatic launcher, with initialization and calibration performed using surface weather conditions measured by the ship's

© Author(s) 2025. CC BY 4.0 License.


onboard instruments. This was done inside an air-conditioned container except for a few launches when the air conditioning was off in order to troubleshoot condensation build-up. Two ground receivers were used during the campaign: one provided by the Max-Planck-Institut für Meteorologie and the other by the German Weather Service (Deutscher Wetterdienst, DWD). Radiosondes launched using the first receiver (for the 00, 06, 12, and 18 UTC ascents) completed their setup on the ground station inside the container. In contrast, those launched using the second receiver (for the 03, 09, 15, and 21 UTC ascents) were initially set up outside for a few minutes before being brought inside for balloon attachment.

Balloons were prepared inside the automatic launcher and filled with Helium, with each balloon inflated until it touched a ball-shaped control element hanging from the ceiling on a string as shown in Figure 3(c). The radiosondes (Vaisala) were initialized using the Vaisala ground station and then carried onto the deck to establish a satellite connection, as the metal container housing the ground station blocked incoming signals. Once the connection was successfully established, the radiosonde was attached to an unwinder, which was then secured to the balloon. Figure 3(c) shows a balloon ready for launch with its radiosonde in the automatic launcher. At the designated launch time, a person had to release a valve to open the launcher, releasing the radiosonde from the port side of the ship. Figure 3 (d) illustrates the open launcher as a radiosonde is launched. During the R/V Meteor deployment, radiosondes were launched every three hours, facilitated by the setup of two Vaisala receivers. Additional radiosondes were occasionally launched between the regular times to coincide with EarthCARE overpasses or flights by the High Altitude and Long Range Research Aircraft (HALO).

Figure 1(c) shows the launch locations of all radiosondes released from the R/V Meteor. The color transitions indicate that the radiosondes drifted farther east of the Atlantic than in the west. The highest height reached was 28.0 km, corresponding to a pressure as low as 16.0 hPa. On average, ascending radiosondes remained airborne for about 167 min and drifted a median distance of 39.5 km, whereas descending radiosondes lasted around 36 min with a median drift of 10.7 km.

Between September 22, 20:30 UTC, and September 24, 03:00 UTC, the R/V Meteor was anchored 1.7 km offshore at 13.16° N, 59.41° W, east of the Barbados Cloud Observatory (BCO) at 13.16° N, 59.43° W (Figure 4 a). This allowed for side-by-side comparisons of overland and overwater soundings. During this period, ten radiosondes were launched simultaneously from both platforms. The average surface wind direction was similar at the two platforms, east-northeast (87.7°) at BCO and east-southeast (91.4°) at the R/V Meteor, indicating a nearly identical flow regime. The mean profiles in Figure 4 (b) and their absolute differences in Figure 4 (c) show very close agreement in both temperature and dew point, with temperature differences remaining well below 1 K. These results support earlier findings (Stevens et al., 2016) that conditions at BCO closely represent those over the adjacent ocean.

## 2.2.1 Oscillatory Radiosonde Ascent During Rain Events

In five cases illustrated in Figure 5, radiosondes launched from the R/V Meteor during rain events displayed pronounced oscillatory behavior. The first ascent phase ended for four of five cases near the freezing level, around 5000 m, after which the radiosonde descended below this altitude before rising again. This ascent–descent cycle repeated several times before the radiosonde ultimately crossed the freezing level and continued to burst altitude. One exception occurred on September 11,

**Figure 4.** Comparison of the ten radiosondes that were coordinated between the BCO and the R/V Meteor. (a) View of the two platforms from Ragged Point Lighthouse (photo credit: Tristan Vostry). (b) Mean Skew-T log-P diagram of the paired soundings. (c) Corresponding absolute differences in temperature and dew point.

Figure 5. Five radiosondes launched during rain events from the R/V Meteor (Vaisala) exhibited oscillatory motion near their respective freezing levels (dashed lines). Rain rates, averaged over the 20 minutes following launch, are shown by the color bar. Measurements were taken by a disdrometer aboard the R/V Meteor and represent conditions at the ship, not along the radiosonde trajectories. The highest rain rate is within the top 2 % of all rainfall recorded during the ship campaign, while even the lowest (yellow) lies within the top 30 %.

2024, at 07:50 UTC, when the initial ascent extended well above the freezing level. The rain rates taken by a disdrometer<sup>10</sup> 130 aboard the R/V Meteor associated with these launches (Figure 5) place all events within the upper third of rainfall intensities measured during the ship deployment, with some among the most intense observed.

One hypothesis is that the balloon became sufficiently moistened during ascent (indicated by the colorbar in Figure 5), allowing ice and snow to form and accumulate on its surface near the freezing point. The added weight of accumulated ice and snow may have temporarily halted the balloon's ascent. As the balloon descended into warmer altitudes, the ice and snow likely melted and ran off, allowing the ascent to resume. Such oscillatory behavior of radiosondes has been reported earlier. Venkiteshwaran (1952) used a radiosonde type with a fan indicating vertical motion of the balloon relative to that of the air. Based on this data it was concluded that two types of cases were observed, some where the descent was presumably due to snow accumulated on the balloon, and others where the descent may have been caused by strong subsidence.

#### 2.3 BCO


On September 7, the first radiosonde was launched from the Barbados Cloud Observatory (BCO, Stevens et al. (2016)), located at 13.16° N, 59.43° W. By September 29, a total of 141 radiosondes were launched, with 135 successful descents recorded.

<sup>&</sup>lt;sup>10</sup>data found at IPFS CID: bafybeihjcwsecgpmsjxoo5peqafnuqfnalu3ya3vtibw17qkm76izsnuei

The radiosondes were prepared inside an insulated container, where the air temperature was maintained at approximately 25°C. Initialization and calibration were conducted using surface weather conditions measured by the BCO instruments.

The balloons were filled outdoors at the platform with Helium using a radiosonde launcher. Figure 3 (e) shows the setup used for this purpose. The radiosonde launcher, positioned in front of BCO to maintain a safe distance from its measuring instruments and fencing (panel f), was covered with a fine cloth to protect the balloon skin and equipped with a holding device for safe handling, even in windy conditions. Each balloon was inflated until it nearly touched the walls of the launcher. A preconfigured radiosonde (Vaisala) fitted with an unwinder was attached to the balloon inside the launcher. At launch time, the balloon was extracted from the radiosonde launcher and released by hand. During the first five days of the BCO ORCESTRA campaign, four radiosondes were launched daily at six-hour intervals. Once the helium supply was secured, the schedule increased to eight radiosondes every three hours on flight days (00, 03, 06, 09, 12, 15, 18, 21 UTC) and six every three hours on non-flight days, excluding the 09 UTC and 21 UTC slots. On the flight days, the higher launch frequency was timed to coincide with the aircraft operations and to enhance the value of joint measurements.

Apart from six post-launch contact losses, no technical issues were encountered during the radiosonde mission at the BCO. Variations in launch times were due to the requirement of obtaining permission from air traffic control at Grantley Adams International Airport, Bridgetown, Barbados for each launch, ensuring no interference with air traffic. As a result, waiting times of up to 30 min were occasionally necessary.

Figure 1 (a) shows trajectory of the radiosonde launched from BCO, with colors indicating height. The radiosondes reached heights of up to  $28.6 \,\mathrm{km}$ , corresponding to pressures as low as  $14.7 \,\mathrm{hPa}$ . On average, ascending radiosondes remained airborne for about  $174 \,\mathrm{min}$  and drifted a median distance of  $24.2 \,\mathrm{km}$ , while the descending radiosondes lasted around  $36 \,\mathrm{min}$  with a median drift of  $8.3 \,\mathrm{km}$ .

#### 3 Data Acquisition




All radiosondes used during the ORCESTRA campaign, whether launched from INMG on Sal Island, the R/V Meteor, or the Barbados Cloud Observatory (BCO), shared several core features. Each radiosonde consisted of a transmitter and battery encased in a protective housing, typically made of styrofoam, and included an external sensor boom to measure temperature and humidity. Humidity sensors were equipped with integrated heating to mitigate the risk of saturation or icing. By design, these sensors measure relative humidity with respect to liquid water, even at freezing temperatures. In addition, all radiosondes carried a Global Positioning System (GPS) receiver, from which horizontal winds were derived. For Vaisala RS41 sondes, the vertical position is available both as a GPS-based altitude referenced to the WGS84 ellipsoid and as a PTU-based height derived from the pressure sensor. In contrast, Meteomodem M20 sondes determine pressure from altitude and thus provide only GPS-based altitude.

At INMG, radiosondes of type M20 from Meteomodem were used (Meteomodem, 2023). These radiosondes descend without a parachute and feature a styrofoam-wrapped casing and sensor boom, as described above. For the radiosonde operations at the R/V Meteor and BCO, systems from Vaisala (RS41-SGP (Vaisala Oyj, 2023) and RS41-SGPe (Vaisala Oyj, 2024)) were


**Figure 6.** Radiosonde launch times plotted on a 24-hour clock. Each radial tick marks a launch, color-coded by platform, with radial distance offset by platform to reduce overlap. Target launch times correspond to standard synoptic and intermediate hours (00, 03, 06, 09, 12, 15, 18, 21 UTC). Most launches occurred 70–80 minutes beforehand to ensure the balloon reached the top of the troposphere near the target hour. Deviations arose from air traffic control restrictions at BCO, adaptive timing aboard the R/V Meteor, and evolving scheduling and technical issues at INMG.

deployed, each equipped with a parachute. The only difference between the two Vaisala models lies in the sensor housing material: the SGP uses styrofoam, while the SGPe employs a biodegradable alternative. The sensors themselves are identical between the two types.

To sample the diurnal cycle of the tropical atmosphere, launches were scheduled so balloons would reach the top of the troposphere near synoptic and intermediate hours (00, 03, 06, 09, 12, 15, 18, 21 UTC). As shown in Figure 6, most launches occurred 70–80 minutes before these target hours. INMG initially operated on an irregular schedule (05:30, 11:30, 14:30, 17:30, 23:30 UTC), but from August 14 shifted to earlier times (04:30, 10:00, 13:30, 16:30, 19:30, 22:30 UTC) to align with data assimilation windows. Although the data were not ultimately assimilated into French NWP models, they were archived for ERA6. Variability in INMG launch times also reflected frequent technical issues and relaunches. The R/V Meteor's dual receiver setup enabled flexible, adaptive launches, including those timed to satellite or airplane overpasses. At BCO, coordination with air traffic control sometimes caused delays relative to the planned schedule.

Figure 7 compares ascent and descent rates for Meteomodem and Vaisala radiosondes. Mean ascent rates (panels (a) and (c)) are stable at around  $5\,\mathrm{m\,s^{-1}}$  for both types, with a spread of approximately  $1\,\mathrm{m\,s^{-1}}$  for Vaisala and  $1.5\,\mathrm{m\,s^{-1}}$  for Meteomodem.

**Figure 7.** Ascent and descent rates of all radiosondes by vendor. Mean values are shown along with the 10th and 90th percentiles and the maximum and minimum rates at each height level. Bold ticks on the x-axes indicate the mean ascent and descent rates below 25 km. The lowest 100 m are omitted because of surface noise.

After balloon burst, mean descent rates in free fall reach up to  $30\,\mathrm{m\,s^{-1}}$ , with a spread of about  $20\,\mathrm{m\,s^{-1}}$  in the upper atmosphere and less than  $5\,\mathrm{m\,s^{-1}}$  near the surface. Below  $15\,\mathrm{km}$ , average ascent and descent rates are  $4.7\,\mathrm{m\,s^{-1}}$  and  $-12.8\,\mathrm{m\,s^{-1}}$  for 190 Meteomodem, and  $4.1\,\mathrm{m\,s^{-1}}$  and  $-10.6\,\mathrm{m\,s^{-1}}$  for Vaisala. The higher mean descent rates of Meteomodem sondes reflect their lack of a parachute, while the most extreme descent values, seen in Vaisala sondes, likely result from parachutes that failed or only partially deployed, increasing effective weight and descent speed (Ingleby et al., 2022). These ascent and descent characteristics are relevant because they determine the temporal resolution of vertical profiles and can influence measurement uncertainty, particularly during descent when conditions for the sensors are less stable.





Finally, Figure S1 shows the mean difference between variables measured during descent and those measured during ascent by the same radiosonde. For Vaisala, the MW41 software introduces a descent-phase height bias (M. Hamouche, Vaisala Oyj, private communication, 2025), so heights were recomputed from GPS altitude by removing the local geoid offset and converting to geopotential height. Each profile was then separated into ascent and descent, pooled, and binned onto a common 10 mgrid before calculating mean differences. Meteomodem data were processed directly using GPS altitude from the Level-1 data, which is unaffected by this issue. Pressure values (panel a) are slightly higher during descent than ascent, with a negligible mean difference of about 12 Pa. Air temperature (panel b) is marginally higher during descent above 15 km and slightly lower during ascent below this height, with mean differences remaining under 0.5 K. A reversed pattern is observed for relative humidity (panel c), with overall mean differences below 0.1 %. The question-mark shape seen for Meteomodem results from sensor time lags that become more pronounced during the faster descent and in very low temperatures near the tropopause. The wind speed (panel d) tends to be higher during ascent than descent, with differences of less than 1 m s<sup>-1</sup>.

To document the processing chain from the raw vendor output to the consolidated campaign dataset, we structure the following description into three stages. First, we describe the Level-0 data, which are the raw radiosonde measurements as delivered by the vendor software. Next, we outline the Level-1 processing, where PySonde is used to harmonize the files, separate ascent and descent, and derive additional thermodynamic variables. Finally, we present the Level-2 dataset, in which mainly the individual profiles are combined and binned onto a common vertical grid.

#### **3.1** Level-0

The data collected by the radiosondes is transmitted every second to the ground station, where vendor-provided software (Meteomodem Eoscan Software and Vaisala Sounding System MW41) processes the incoming signal and applies an initial quality control prior to any further processing. Both systems detect transmission errors, apply plausibility checks, and record missing or interpolated values with placeholder values.

For Vaisala RS41 radiosondes, this process begins with a mandatory ground check prior to launch. The factory calibration is restored, humidity sensors are reconditioned and subjected to a zero-humidity check, and a built-in temperature check detects damaged units before flight. Pressure sensors are calibrated against a barometer module in the ground check device. During flight, the MW41 software continuously monitors pressure, temperature, humidity, and wind components (zonal, meridional, and absolute). Horizontal winds and positions are derived from GPS signals, while vertical position is available either from a pressure sensor or from GPS-derived altitude, referenced to the WGS84 ellipsoid (Vaisala, 2013).

For Meteomodem M20 radiosondes, the vendor software performs comparable pre-flight checks, verifying sensor performance against surface reference values, restoring calibration, and reconditioning humidity sensors. In flight, the software applies plausibility checks and manufacturer-specific humidity corrections (Dupont et al., 2020), including adjustments for sensor lag and temperature-dependent response. Pressure is determined from GPS altitude, complemented by an onboard barometer near the surface for improved accuracy.

The resulting vendor "raw" data, having passed the mentioned vendor's preliminary tests, constitutes the Level-0 product. The Meteomodem software generates files in .gps format, which are subsequently processed into .cor files, while Vaisala's

Table 2. Summary of processing levels, variables, and quality control applied to the ORCESTRA radiosonde datasets.

| Level                                   | Variables Contained                                                                                                                                                                                                                                                                                                                                                                                         | <b>Processing Steps</b>                                                                                                                                                                                                                                                                                                                                 | Quality Control (QC)                                                                                                                                                                                          |  |
|-----------------------------------------|-------------------------------------------------------------------------------------------------------------------------------------------------------------------------------------------------------------------------------------------------------------------------------------------------------------------------------------------------------------------------------------------------------------|---------------------------------------------------------------------------------------------------------------------------------------------------------------------------------------------------------------------------------------------------------------------------------------------------------------------------------------------------------|---------------------------------------------------------------------------------------------------------------------------------------------------------------------------------------------------------------|--|
| Level-0                                 | <b>Data variables:</b> pressure, temperature, relative humidity, wind speed and direction, horizontal and vertical position, and metadata.                                                                                                                                                                                                                                                                  | Data transmitted every second to ground station; vendor software (MW41 / Eoscan) writes raw files (.mwx for Vaisala, .cor for Meteomodem). Minimal vendor preprocessing: ground check, calibration restore, humidity reconditioning, burst detection.                                                                                                   | Vendor QC: transmission error detection, plausibility checks, basic temperature and humidity correction (including radiation error adjustment), flagging with placeholders.                                   |  |
| Level-1                                 | Dimensions: sonde_id × level.  Coordinates: lat (°N), lon (°E), flight_time (UTC), launch_time (UTC), level, sonde_id.  Data variables: alt (m; altitude above WGS84 ellipsoid), p (Pa), ta (K), rh (1), dp (K), mr (1), dz (m s <sup>-1</sup> ; ascent/descent rate), height (m; geopotential height from PTU), wspd (m s <sup>-1</sup> ), wdir (deg), platform (label), launch_lat (°N), launch_lon (°E). | Level-0 parsed and harmonized with PySonde; placeholders converted to NaN; units/standard_names set; profiles split into ascent/descent (one <b>NetCDF</b> file per branch); derived variables computed: $dp$ , $mr$ , $dz$ ; GPS altitude always used for $alt$ , PTU height retrieved from Vaisala .mwx files and computed for Meteomodem .cor files. | Structural checks (monotonicity of altitude within each branch, duplicate levels removed); physically impossible vendor values already masked as NaN; nonmonotonic sections retained but flagged in metadata. |  |
| Level-1<br>(Oscillating<br>Radiosondes) | Dimensions: sonde_id × sample.  Coordinates: lat (°N), lon (°E), flight_time  (UTC), release_time (UTC), sample,  sonde_id. Data variables: p (Pa), alt  (m; altitude above WGS84 ellipsoid), ta  (K), rh (1), dp (K), mr (1), dz (m s <sup>-1</sup> ),  wspd (m s <sup>-1</sup> ), wdir (deg), platform (label),  launch_lat (°N), launch_lon (°E).                                                        | Raw .xml files inside .mwx archives parsed directly, bypassing MW41 burst-cropping. No ascent/descent split; whole oscillatory trajectory kept in one profile. All five oscillating soundings concatenated into a dedicated Level-1 .zarr dataset.                                                                                                      | QC restricted to NaN handling and thermodynamic plausibility checks; monotonicity of altitude not applicable. No Level-2 product created due to oscillatory vertical profiles.                                |  |
| Level-2                                 | <b>Dimensions:</b> launch_time $\times$ height. <b>Coordinates:</b> height (m), interpolated_time (UTC), launch_time (UTC), lat (°N), lon (°E). <b>Data variables:</b> all Level-1 variables, plus $\theta$ (K), $q$ (kg kg $^{-1}$ ), wind components $u, v$ (m s $^{-1}$ ), iwv (kg m $^{-2}$ ), and launch position metadata.                                                                            | All profiles concatenated into a campaign-wide dataset; all variables vertically interpolated to a uniform 10 m grid up to 31 km. Interpolation combines linear height and log-pressure schemes, masked by occupancy to preserve data gaps. Stored as .zarr dataset.                                                                                    | QC ensures structural consistency: height monotonicity enforced, duplicates removed, interpolated values masked by occupancy to retain data gaps. No smoothing or bias correction applied.                    |  |





MW41 software produces .mwx files, which are compressed directories containing plain-text .xml files with the measured variables. These outputs provide the highest-resolution record of the radiosonde transmission and serve as the starting point for Level-1 processing. An overview of the variables, processing steps, and quality control across all levels is summarized in Table 2.

## 3.1.1 Quality Control at Level-0

In both systems, the vendor software provides the first layer of quality control (QC) by flagging or removing implausible values, correcting for known sensor biases (such as humidity response times), and monitoring data continuity throughout the sounding. This automated QC ensures that Level-0 outputs already meet a baseline reliability before further reprocessing.

#### 3.2 Level-1

We use the PySonde package v0.0.7 (Schulz et al., 2024) to restructure the quality-controlled Level-0 files into consistent single-profile ascent and descent datasets, to add derived variables, and to save the Level-1 output in NetCDF format. An earlier version of this package was employed during the EUREC<sup>4</sup>A campaign to process radiosonde data (Stephan et al., 2021). We have extended PySonde to process .cor files, enabling us to create a comparable dataset that facilitates easy comparisons between radiosonde measurements from EUREC<sup>4</sup>A and ORCESTRA (see Table 2).

In the first step, PySonde extracts measured values and their associated units from .cor and .mwx files. Vendor-specific placeholder values for missing data are systematically converted to NaN to ensure consistent handling of gaps across platforms. Next, the radiosonde profile is split into ascending and descending segments based on the balloon burst. For Vaisala data, the MW41 software provides an internal burst flag when it detects a sustained increase in pressure; for Meteomodem data, PySonde identifies bursts from the transition in the altitude–time series.

For the Level-1 product, several additional variables are derived. The ascent rate is computed from successive height–time differences. Dew-point temperature is calculated from measured temperature T and relative humidity rh following the empirical Vaisala formulation,

$$T_d = \frac{2T\Lambda}{T\ln\left(\frac{100}{\text{rh}}\right) + 2\Lambda}, \qquad \Lambda = 15\ln\left(\frac{100}{\text{rh}}\right) - 2(T - 273.15) + 2711.5,\tag{1}$$

where  $\Lambda$  is an empirical function that approximates the Magnus-type relation between temperature, relative humidity, and dew point used in the Vaisala MW41 system. The saturation mixing ratio is derived following standard thermodynamics (see below). Launch times are standardized to UTC datetimes and stored as the coordinate launch\_time, ensuring consistent indexing of soundings across platforms. The resulting data are then structured into NetCDF files, saved separately for the ascending and descending branches. Each file in the Level-1 product contains two dimensions: launch\_time, indexing the individual radiosonde profile, and level, representing the vertical sampling points along each ascent or descent.

The mixing ratio mr is computed via:

$$mr = \frac{\epsilon \cdot rh \cdot e_s}{p - rh \cdot e_s},\tag{2}$$

with  $e_{\rm s}$  the saturation vapor pressure of water, rh the relative humidity, p is the atmospheric pressure and  $\epsilon = \frac{M_v}{M_d}$  is the ratio of the molar mass of water vapor to that of dry air.

We compute the saturation vapor pressure of water  $e_s$  according to the IAPWS (International Association for the Properties of Water and Steam) formulation by Wagner and Pruß (2002):

$$e_s(T) = P_c \exp\left(\frac{T_c}{T} \left(a_1 v_t + a_2 v_t^{1.5} + a_3 v_t^3 + a_4 v_t^{3.5} + a_5 v_t^4 + a_6 v_t^{7.5}\right)\right)$$
(3)

with T the air temperature,  $T_c$  the critical temperature of water vapor,  $P_c$  critical pressure of water vapor,  $v_t = 1 - TT_c^{-1}$  and the empirical coefficients:  $a_1 = -7.85951783$ ,  $a_2 = 1.84408259$ ,  $a_3 = -11.7866497$ ,  $a_4 = 22.6807411$ ,  $a_5 = -15.9618719$ ,  $a_6 = 1.80122502$ .

#### 3.2.1 Quality Control at Level-1

In addition to the vendor checks already applied at Level-0, PySonde performs additional quality control. All vendor place270 holders are replaced by NaN, and values outside physically reasonable ranges (e.g. negative pressures, relative humidities 
> 100%, or temperatures < 150 K and > 330 K) are masked. Profiles are checked for monotonicity of height during ascent 
and descent; non-monotonic sections are retained but flagged, as they imply that ascent rate estimates are not physically meaningful. Duplicate vertical levels are removed to avoid inconsistencies in later binning. The guiding principle of Level-1 QC is 
therefore to ensure structural consistency and remove physically implausible values, while preserving the original measurement 
information as completely as possible.

# 3.2.2 Oscillating Radiosondes

As described in Section 2.2.1 and shown in Figure 5, five Vaisala RS41 soundings from R/V Meteor exhibited oscillatory motion during ascent. Because the MW41 software misclassified the first temporary descent as the balloon burst, the Level-0 files were prematurely truncated. To recover the full data, we directly accessed the unprocessed .xml files contained inside the .mwx archives. These .xml files provide the actual raw measurements from Vaisala, meaning that the standard vendor QC applied at Level-0 was not included here. In contrast to the standard workflow, these trajectories were not split into separate ascent and descent branches, since such a distinction is not meaningful for oscillating profiles. Instead, each trajectory was retained in its entirety as a single file with the dimensions sounding × sample

From these raw measurements we extracted the standard observed variables and, using individual PySonde functions, computed dew point temperature and mixing ratio. The five recovered soundings were merged into a dedicated Level-1 .zarr dataset. No Level-2 product was generated for the oscillating radiosondes (see Table 2), as Level-2 processing requires monotonic ascent or descent and subsequent binning onto a uniform vertical grid, conditions not met by these oscillatory profiles.

# 3.3 Level-2

280

290

The Level-2 product builds on the processed Level-1 data (cf. Table 2), which contain the cleaned and derived measurements from individual soundings.

Before vertical interpolation, each Level-1 profile is checked for monotonicity in its height coordinate. For both ascent and descent segments, height is expected to vary consistently in one direction. Duplicate height levels are removed to ensure uniqueness.

Each profile is then vertically interpolated onto a uniform height grid extending from the launch level to  $31 \, \mathrm{km}$  in  $10 \, \mathrm{m}$  increments. The vertical interpolation combines a linear interpolation in height with a dedicated log-pressure interpolation to preserve the physical structure of the p(z) profile. A per-variable occupancy mask, derived from a temporary bin, ensures that interpolated values are retained only where the original profile contained valid data, preventing artificial layers. This two-step approach preserves the vertical integrity of the measurements while maintaining data honesty by leaving gaps where no observations exist in the raw profile.

On this common vertical grid, additional variables are derived, including potential temperature ( $\theta$ ), specific humidity (q), and integrated water vapour (IWV). After interpolation and variable derivation, all processed profiles are concatenated into a combined multi-sounding dataset and stored in Zarr format for distribution via the InterPlanetary File System (IPFS).

We derive potential temperature, specific humidity and IWV as follows. Potential temperature is computed with MetPy as

$$\theta = T \left(\frac{p_0}{p}\right)^{R_d/c_{p_d}}, \qquad p_0 = 10^5 \text{ Pa},$$
(4)

where T is air temperature (K), p is pressure (Pa),  $R_d$  is the gas constant for dry air, and  $c_{pd}$  is the specific heat of dry air at constant pressure.

Specific humidity q is obtained from the mixing ratio mr via

$$q = \frac{\mathrm{mr}}{1 + \mathrm{mr}}.$$
 (5)

IWV is computed by vertically integrating the water-vapour mass density over height. For this, realistic values require continuous measurements, particularly in the lower atmosphere where most of the water vapor resides. Profiles with missing data segments would lead to biased or physically meaningless IWV estimates. Therefore, we apply a four-step filtering procedure before solving the following integral (Schulz and Stevens, 2018):

$$IWV = \int q(z) \rho(z) dz, \tag{6}$$

where q is the specific humidity and  $\rho$  the air density derived from pressure and virtual temperature.

First, we exclude soundings from R/V Meteor launched before 16 August, when the vessel was still docked in the port of Mindelo, Cape Verde, and thus did not sample open-ocean conditions.

Second, we retain soundings that reach at least 8 km height and contain data below this level. The 8 km threshold is a practical cutoff chosen for comparability with HALO dropsonde coverage during PERCUSION, although many dropsonde profiles extend higher.

Third, within the 0-8 km layer we require that no more than 20% of vertical levels are missing when considering the simultaneous availability of humidity q, pressure p, and temperature T (i.e., a level counts as valid only if all three are finite).

© Author(s) 2025. CC BY 4.0 License.

Science Science Data

Fourth, we enforce sufficient near-surface sampling by keeping only profiles with at least 50 levels in the lowest 1 km where q, p, and T are all finite.

Small internal gaps are then filled along height using linear interpolation for q and T, and log-linear interpolation for p; endpoints of p and T are extrapolated by forward/backward filling. Near-surface q is backfilled only up to  $300\,\mathrm{m}$  to bridge short launch-adjacent gaps; if the first (surface) level of q remains missing after this limited backfill (e.g., a gap  $> 300\,\mathrm{m}$ ), the profile's IWV is masked and excluded from IWV statistics. We do not extrapolate q beyond this limit (only p and T are extrapolated at the ends). In total,  $72.1\,\%$  of soundings remain available for analysis after filtering.

#### 3.3.1 Quality Control at Level-2

Quality control at Level-2 focuses on ensuring structural consistency of the vertical profiles and transparency of the binning procedure. Empty grid points remain explicitly missing rather than filled by interpolation, and duplicate or non-monotonic levels are removed. No additional smoothing or bias correction is applied: the guiding principle of Level-2 QC is to provide a harmonized, intercomparable dataset while preserving the raw variability of individual soundings.

### 4 Observed Atmospheric Conditions During ORCESTRA

Figure 8 summarizes the vertical structure of winds, humidity, and moist static energy observed during ORCESTRA. Across all sites, the profiles reveal coherent patterns that reflect the large-scale circulation of the tropical Atlantic rather than local variability. Easterly winds dominate the zonal wind profiles at BCO, INMG, and the R/V Meteor west of 40°W, while the R/V Meteor east of 40°W records westerlies below 700 hPa that shift to easterlies aloft. In the meridional wind, BCO and the R/V Meteor show northerlies below 700 hPa, whereas INMG exhibits southerlies in the lower atmosphere and northerlies above, resulting in a mean net northward flow.

Relative humidity in the boundary layer is highest at INMG and on the R/V Meteor east of 40°W, followed by the R/V Meteor west of 40°W and BCO. At INMG, humidity decreases sharply just above the surface until about 700 hPa, while at the other platforms the decline with height is more gradual.

Moist static energy (MSE) was computed following  $MSE = c_p T + L_v q + gz$ , where T is temperature, q specific humidity, and z height. The calculation used the Python package moist\_thermodynamics by Bjorn Stevens<sup>11</sup>, which incorporates composition-dependent heat capacities. Relative humidity was recomputed from radiosonde measurements using temperature-dependent saturation vapor pressures over liquid and ice, based on the IAPWS formulations of Wagner and Pruß (2002) for liquid water and Wagner et al. (2011) for ice. The highest MSE values in the lower atmosphere are observed at BCO, followed by the R/V Meteor west of  $40^{\circ}$ W, R/V Meteor east of  $40^{\circ}$ W, and INMG. Above  $700 \, \text{hPa}$ , MSE from BCO is the smallest, while the R/V Meteor east of  $40^{\circ}$ W shows the slowest decrease with height.

Figure 9 shows the evolution of vertical relative humidity profiles throughout the campaign. The R/V Meteor's early profiles, collected in the eastern Atlantic near Sal (22.56°W), closely resemble those from INMG, while later profiles, taken when the



<sup>&</sup>lt;sup>11</sup>https://github.com/mpimet/moist\_thermodynamics



Figure 8. Mean profiles from the entire ORCESTRA campaign, based on radiosonde data from BCO, INMG (Sal Island), and the R/V Meteor. Shipboard profiles are split into two regions: east of  $40^{\circ}$ W and west of  $40^{\circ}$ W, representing different phases of the Atlantic transect. Panels show (a) zonal wind, (b) meridional wind, (c) relative humidity (computed over liquid above and over ice below freezing temperature), and (d) moist static energy as a function of pressure. Profiles are averaged over both ascending and descending soundings. Markers at the top indicate the mean (circle) and one standard deviation (horizontal bars) of the column-averaged values for each platform.

ship was anchored near BCO (59.43°W), match BCO's vertical structure more closely. Relative humidity was recalculated over liquid water for T>0°C and over ice for T

**Figure 9.** Time—height cross sections of relative humidity from radiosondes launched at (a) INMG on Sal Island, (b) the R/V Meteor, and (c) BCO, covering the full campaign period at each platform. In panel (b), a secondary y-axis at the top shows the ship's longitude as it traversed the tropical Atlantic from east to west.

**Figure 10.** Histograms of integrated water vapor (IWV) from radiosonde launches at BCO, the R/V Meteor, and INMG. Bold vertical ticks along the x-axis indicate the mean IWV for each platform. Analysis includes 72.1 % of all soundings after filtering.

after the peak wind speed of  $42.1\,\mathrm{m\,s^{-1}}$  at  $13.5\,\mathrm{km}$ . The coldest overall temperature,  $-85.5\,^{\circ}\mathrm{C}$ , was recorded by a radiosonde launched from INMG on August 24 at 22:33 at  $17.5\,\mathrm{km}$ .

Figure S3 shows the maximum height reached by each radiosonde per platform as a histogram. The campaign median is 25.1 km indicated by the dashed black line. Radiosondes launched from INMG on Sal Island, Cape Verde, show a small peak below 5 km which can be linked to early loss of contact with radiosondes immediately after launch in the beginning of the campaign, caused by an interference signal. While the radiosondes launched from BCO (Vaisala, RS41-SGP&-SGPe) reached highest height values, the majority of all radiosondes launched from R/V Meteor (Vaisala, RS41-SGP&-SGPe) climbed further than the campaign median. Radiosondes launched from INMG (Meteomodem) have a tendency to burst before the campaign median.

#### 5 Code and data availability



The raw data output by the respective software of the radiosonde manufacturers Vaisala and Meteomodem after a successful measurement was processed using PySonde (v0.0.7) (Schulz et al., 2024), converting Level-0 data to Level-1 and subsequently to Level-2. For five oscillating Vaisala RS41 sondes launched from R/V Meteor, the manufacturer's software prematurely truncated the profiles; these were therefore recovered directly from the unprocessed .xml files within the .mwx archives.

Table 3. Selected extreme values measured by radiosondes at each platform, including UTC timestamps and corresponding heights. The five oscillating radiosondes launched from the R/V Meteor are excluded.

|            | Coldest<br>Temperature                               | Max. Wind<br>Speed                                                | Min. Wind<br>Speed                                               | Longest<br>Distance                    | Highest<br>Height                     | Lowest<br>Pressure                                   |
|------------|------------------------------------------------------|-------------------------------------------------------------------|------------------------------------------------------------------|----------------------------------------|---------------------------------------|------------------------------------------------------|
| ВСО        | -80.8 °C<br>(2024-09-09T<br>16:48:39Z<br>at 16.6 km) | 29.4 ms <sup>-1</sup><br>(2024-09-13T<br>22:52:28Z<br>at 26.4 km) | 0.0 m s <sup>-1</sup><br>(2024-09-12T<br>19:49:59Z<br>at 6.5 km) | 62.2 km<br>(2024-09-14T<br>19:53:38Z)  | 28.6 km<br>(2024-09-14T<br>19:53:38Z) | 14.7 hPa<br>(2024-09-14T<br>19:53:38Z<br>at 28.5 km) |
| R/V Meteor | -84.9 °C<br>(2024-08-24T<br>19:50:05Z<br>at 17.7 km) | 42.1 ms <sup>-1</sup><br>(2024-08-24T<br>10:46:17Z<br>at 13.4 km) | $0.0 \mathrm{ms^{-1}}$ (2024-09-23T 00:39:17Z at 17.2 km)        | 116.9 km<br>(2024-08-25T<br>16:50:16Z) | 28.0 km<br>(2024-09-12T<br>22:47:02Z) | 16.0 hPa<br>(2024-09-12T<br>22:47:02Z<br>at 28.0 km) |
| INMG       | -85.5 °C<br>(2024-08-24T<br>22:33:52Z<br>at 17.5 km) | $32.8\mathrm{ms^{-1}}$ (2024-08-22T 20:50:38Z at 20.6 km)         | 0.0 m s <sup>-1</sup><br>(2024-09-05T<br>23:49:51Z<br>at 4.8 km) | 89.8 km<br>(2024-08-15T<br>17:24:59Z)  | 27.4 km<br>(2024-08-14T<br>14:29:08Z) | 17.5 hPa<br>(2024-08-14T<br>14:29:08Z<br>at 27.3 km) |

The resulting datasets are available via IPFS and can be explored using the ORCESTRA Data Browser<sup>12</sup> or accessed directly via IPFS. The following datasets were produced as part of this study (Winkler et al., 2025a): 380

# - RAPSODI Raw Radiosonde Measurements during ORCESTRA (Level 0):

BCO (Winkler et al., 2025b): https://latest.orcestra-campaign.org/raw/BCO/radiosondes/ INMG (Winkler et al., 2025c): https://latest.orcestra-campaign.org/raw/INMG/radiosondes/ R/V Meteor (Winkler et al., 2025d): https://latest.orcestra-campaign.org/raw/METEOR/radiosondes/

#### 385 - RAPSODI Oscillating Radiosonde Measurements during ORCESTRA (Level 1):

https://browser.orcestra-campaign.org/ Paste name of the dataset into the search window. (Winkler et al., 2025e)

# - RAPSODI Radiosonde Measurements during ORCESTRA (Level 1) (merged, padded to common vertical levels):

https://browser.orcestra-campaign.org/

Paste name of the dataset into the search window.

(Winkler and Rixen, 2025a)

<sup>12</sup>https://browser.orcestra-campaign.org


# - RAPSODI Radiosonde Measurements during ORCESTRA (Level 2):

https://browser.orcestra-campaign.org/

Paste name of the dataset into the search window.

(Winkler and Rixen, 2025b)

Datasets can be accessed through any web browser, although installing and running an IPFS client is recommended for direct data retrieval.<sup>13</sup> The ORCESTRA Data Browser also provides a graphical interface to search and preview the zarr-dataset collections.

The Python scripts used to generate all figures in this paper are available in a dedicated repository<sup>14</sup>. The version of PySonde employed for the processing, including campaign-specific configuration files, is archived separately <sup>15</sup>.

In Python, the data can be accessed directly from IPFS. The following example shows how to load the Level-1 oscillating radiosonde, Level-1 merged (padded to common vertical levels), and Level-2 concatenated datasets:

```
       ı: import xarray as xr
       3: RAPSODI_Oscillating = xr.open_dataset(
             "ipfs://bafybeidtfcyurvbw5obbhl5zlyfaoamnlhu6rrxyrmg2r2wwwidxg32oeg",
             engine="zarr"
       7: RAPSODI_Level1 = xr.open_dataset(
             "ipfs://bafybeidol5jadpzb2ibssf2lbbgdhm2zgzeg2urdyefwsxx7eelmcuumn4",
             engine="zarr"
      10: )
      ii: RAPSODI_Level2 = xr.open_dataset(
             "ipfs://bafybeid7cnw62zmzfqxcvc6q6fa267a7ivk2wcchbmkoyk4kdi5z2yj2w4",
      12:
             engine="zarr"
      13:
      14: )
```

#### 420 6 Summary


This paper presents the Radiosonde Atmospheric Profiles from Ship and island platforms during ORCESTRA, collected to Decipher the ITCZ (RAPSODI) — a quality-controlled dataset encompassing all radiosonde profiles conducted during the ORCESTRA campaign. ORCESTRA is a large-scale field study carried out in August and September 2024, with radiosonde observations playing a central role. These observations provide vertical profiles of temperature, humidity, wind, and pressure across the tropical Atlantic, spanning the region between 4°N and 17°N, and from 20°W to 60°W.

Radiosondes were launched from three platforms strategically distributed across the basin: the National Institute of Meteorology and Geophysics (INMG) on Sal Island, Cape Verde, in the eastern Atlantic; the research vessel R/V Meteor, which

<sup>13</sup> https://docs.ipfs.tech/concepts/what-is-ipfs/

<sup>&</sup>lt;sup>14</sup>https://github.com/mariuswinkler/Winkler\_et\_al\_RAPSODI\_Data\_Paper\_2025.git

<sup>&</sup>lt;sup>15</sup>https://github.com/mariuswinkler/Winkler\_et\_al\_RAPSODI\_PySonde\_for\_Data\_Paper\_2025.git

© Author(s) 2025. CC BY 4.0 License.





Science Science Data

traversed the Atlantic from east to west; and the Barbados Cloud Observatory (BCO) in the western Atlantic. Over the course of the campaign, a total of 624 radiosondes were launched, most of which returned both ascent and descent measurements. Data quality varied across platforms due to factors such as early loss caused by radio interference and environmental conditions, including humidity and precipitation.

The dataset was processed in three stages. Level-0 comprises the raw data output by the Meteomodem and Vaisala ground station software, which already includes the manufacturer's built-in quality control such as ground checks, calibration restoration, humidity sensor reconditioning, and plausibility filtering during flight. Level-1 data is generated from these files using the PySonde package, which restructures the profiles into consistent ascent and descent segments, converts vendor placeholders to missing values, and adds several derived thermodynamic variables. In addition, a dedicated Level-1 dataset was created for five oscillating Vaisala sondes launched from R/V Meteor, for which PySonde could not be applied. Level-2 builds on the Level-1 product by checking for monotonic ascent or descent, computing further diagnostics, and binning all soundings onto a uniform vertical grid for analysis.

Radiosonde profiles were collected across the Atlantic Intertropical Convergence Zone (ITCZ), providing coverage of the atmospheric structure along an east—west transect. The vertical profiles reflect a gradient in atmospheric moisture: R/V Meteor operated predominantly in moist conditions, BCO sampled both moist and dry environments, and INMG remained primarily in drier air throughout the campaign. In particularly moist conditions associated with heavy rainfall, some profiles from R/V Meteor exhibited rare oscillatory behavior. The quality-controlled dataset is published in Zarr format on the Inter Planetary File System (IPFS) for public access.

### Acknowledgments

We thank the staff of the INMG and the Mindelo Ocean Science Center in Cape Verde for their collaboration and operational support. The atmospheric soundings at INMG benefited from access to the National Instrument Moyens Mobiles, which is part of the ACTRIS-FR research infrastructure.

We are grateful to the Deutscher Wetterdienst (DWD) for providing the second receiver on R/V Meteor. Martin Stelzner (DWD) contributed one launch per day as part of the regular operation of the DWD weather station on the ship and provided essential advice on launch procedures at sea. Nearly all members of the BOWTIE science team as well as the crew of Meteor assisted with radiosonde launches, without whom the campaign would not have been possible. Technical support for the installation and deinstallation phases on Meteor was provided by Friedhelm Jansen and Björn Brüggemann.

At BCO, radiosonde operations and infrastructure were supported by the technical staff Friedhelm Jansen and Björn Brüggemann. Radiosondes at the BCO were in part funded through the CLICCS Cluster of Excellence (EXC 2037, Project 390683824).

We gratefully acknowledge Lukas Kluft and Tobias Koelling for their support in establishing and maintaining the IPFS infrastructure, and thank Theresa Mieslinger for her constructive internal review.

Funding for R/V Meteor cruise M203 was provided by the Deutsche Forschungsgemeinschaft (DFG) under grant number 460 GPF20-1 072. Raphaela Vogel and Nina Robbins-Blanch acknowledge support from the European Research Council (ERC)

© Author(s) 2025. CC BY 4.0 License.

Science Science Data

under the Horizon Europe programme (ROTOR, grant no. 101116282). Fleur Couvreux, Florent Beucher, and Philippe Peyrillé acknowledge support from LEFE (ILSOM project). PICCOLO and its participants (MMB, DCB, WTH, SK, MK, TYL, ML, EL, NRM, JHR, JCS, CS, MS, AT, CW, and AAW) are supported by the U.S. National Science Foundation (NSF) through Awards No. 2331199, 2331200, and 2331202.

#### 465 Author Contributions and Ordering

Writing and visualization (original draft) were carried out by MW, with plotting contributions from MR. Writing (review and editing) and dataset validation were performed by MW with feedback from MR, FB, FC, PP, HS, HS, JC, EF, HMG, BP, NRB, DK, RV, SB, AAW, and BS.

Data curation included adapting the PySonde package to campaign needs, with MR implementing the processing of proprietary .cor files. KHW developed a method to process oscillating soundings directly from raw .mwx files, bypassing the MW41 system limitations. MW led the overall data curation, processing, and quality control, and ensured persistent dataset availability through the IPFS. The author list reflects different levels of responsibility and contribution to the radiosonde operations and dataset preparation.

# Group 1 — Mission leads (MW-KHW).

MW and MR were responsible for BCO; FB, FC, and PP for INMG; HS, HS, and KHW (together with DK and AAW from Group 3) for R/V Meteor. Tasks included ensuring that hardware and software were in place, creating launch schedules, managing shifts, and taking overall responsibility in coordination with the principal investigators (Group 3). CCN contributed to scheduling FSU students who helped launch radiosondes at BCO.

#### Group 2 — Launch operations (EAB-CW).

(Indicated by a balloon symbol in the author list.) These authors participated in radiosonde shifts and took responsibility for timely launches. Authors in this group are listed alphabetically.

#### Group 3 — Strategy (DK-BS).

Principal investigators who contributed to the design of the overall radiosonde strategy.

# **Competing Interests**

The authors declare that they have no conflict of interest.

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
