# Peer review of "RAPSODI: Radiosonde Atmospheric Profiles from Ship and island platforms during ORCESTRA, collected to Decipher the ITCZ"

_Earth System Science Data, 2025_

## Referee Comment (RC1)

Review of "RAPSODI: Radiosonde Atmospheric Profiles from Ship and Island platforms during ORCESTRA, collected to Decipher the ITCZ" by Winkler et al. (2026).

This dataset, conducted by a collaborative team over ocean and island platforms, presents a potentially valuable resource for investigating fine-scale vertical atmospheric structures. The experiment itself appears both well-designed and scientifically engaging. Compared to other ocean-based sounding campaigns, the launch frequency in this study is notably higher in temporal density, which should offer more detailed perspectives for studying the Intertropical Convergence Zone (ITCZ). Overall, the paper is well-organized and provides thorough technical documentation. This research holds great potential, and the dataset itself possesses significant scientific value. I have benefited from the authors' academic rigor and would like to express my appreciation for their thoughtful work. I would recommend major revision to the manuscript prior to publication.

**Comments:**

1. As a data description paper, I found it quite difficult to access the data link provided in the Abstract. Many readers may not be familiar with the IPFS system. Providing an alternative HTTP address would be more appropriate. Since the author provides corresponding Python code later in the text, it might be better to include a brief explanation in the abstract.

2. To my knowledge, there have been many other Atlantic radiosonde launch campaigns, such as POLARSTERN, DBLK, HTXUH4H, among others. On an old hard drive, I discovered approximately 20,000 high-resolution radiosonde profiles launched over the ocean, most of which were conducted over the Atlantic. As an example, I have provided two screenshots of the data listing below. Although I have not systematically examined the spatial overlap between these data and the study area, it might be worthwhile to briefly introduce other oceanic radiosonde experiments in the introduction and provide a concise comparison. This would allow readers to gain a broader perspective on the full scope of Atlantic radiosonde campaigns.

[Figure]

3. The authors emphasize the ITCZ in their title, yet more detailed analysis of the ITCZ is not found in the main text. If feasible, providing a preliminary finding on the ITCZ could enhance the scientific contribution of this paper.

4. The abstract should include more details about the radiosonde dataset, such as the time range, release intervals, and balloon sampling frequency. Additionally, the R/V Meteor may be difficult to understand for readers unfamiliar with German scientific expeditions; for instance, I initially assumed it referred to a type of meteor radar.

5. The Introduction would benefit from a clearer explanation of the scientific motivation behind ORCESTRA. What were the key research questions or atmospheric processes that this campaign aimed to address?

6. It is recommended that the following content from page 2 be annotated within the main text. This citation format appears inconsistent with standard EGU citation style and caused some confusion. Additionally, page 3 contains similar phrasing such as "MAESTRO (mesoscale organization of tropical convection)," which duplicates the expression on page two.
* * *
[1] https://orcestra-campaign.org/orcestra.html
[2] PERCUSION ≡ Persistent EarthCare underflight studies of the ITCZ and organized convection
[3] MAESTRO ≡ Mesoscale organisation of tropical convection
[4] BOWTIE ≡ Beobachtung von Ozean und Wolken – Das Trans ITCZ Experiment
[5] PICCOLO ≡ Process Investigation of Clouds and Convective Organization over the atLantic Ocean
[6] SCORE ≡ Sub-Cloud Observations of Rain Evaporation

7.  Figure 1a: Why do some trajectory segments appear discontinuous in the lower-right part of the panel? This is unusual in my experience—could it be due to data loss? For panels (a) and (b), which represent land-based ("stationary") platforms, adding launch coordinates to the figure would be helpful. Additionally, I made every effort to interpret Figure 1, as it is crucial for understanding the entire experiment. Unfortunately, despite over a decade of radiosonde experience, I find Figure 1 difficult to comprehend. The phrase "...both ascending and descending segments shown" is perplexing. Typically, descending balloons, as described in the main text, are called dropsondes, while ascending ones are radiosondes. I'm unclear on what exactly the "descending" values represent. Does "descending" refer to the period after balloon burst? A typical radiosonde profile (sampling rate = 1s) would produce a continuous curve rather than the scattered points shown in (b).

8.  Given that Meteomodem and Vaisala radiosondes are well-established and widely documented technologies (e.g., in journals like AMT), the authors might consider reducing the technical details in favor of highlighting the unique scientific opportunities offered by the ORCESTRA campaign. What are the potential research themes enabled by this dataset? Which atmospheric processes could be better examined? Expanding the Summary to include such perspectives would increase the impact and value of the data.

---

## Author Comment (AC1)

In review:

**RAPSODI: Radiosonde Atmospheric Profiles from Ship and island platforms during ORCESTRA, collected to Decipher the ITCZ**

Marius Winkler, Marius Rixen, Florent Beucher, Fleur Couvreux, Chaehyeon C. Nam, Philippe Peyrillé, Hauke Schmidt, Hans Segura, Karl-Hermann Wieners, Ezri Alkilani-Brown, Abdou Aziz Coly, Giovanni Biagioli, Michael M. Bell, Ester Brito, Emma Chauvin, Julie Capo, Delián Colón-Burgos, Akeem Dawes, Jose Carlos da Luz, Zekican Demiralay, Vincent Douet, Vincent Ducastin, Clarisse Dufaux, Jean-Louis Dufresne, Florence Favot, Thomas Fiolleau, Emilie Fons, Geet George, Helene M. Gloeckner, Suelly Gonçalves, Laurent Gouttesoulard, Lennéa Hayo, Wei-Ting Hsiao, Sarah Kennison, Michael Kopelman, Tsung-Yung Lee, Enora Le Gall, Mateo Lovato, Emily Luschen, Nicolas Maury, Brett McKim, Louis Netz, Diouf Ousseynou, Karsten Peters-von Gehlen, Chavez Pope, Basile Poujol, Niwde Rivera Maldonado, Nina Robbins Blanch, Nicolas Rochetin, Daniel Rowe, Paula Romero Jure, James H. Ruppert Jr., Jairo Segura Bermudez, Jarrett C. Starr, Martin Stelzner, Connor Stoll, Macintyre Syrett, Abraham Tekoe, Jeremie Trules, Colin Welty, Daniel Klocke, Raphaela Vogel, Sandrine Bony, Allison A. Wing, Bjorn Stevens

February 7, 2026

**Response to RC-1[1]**

*Review of "RAPSODI: Radiosonde Atmospheric Profiles from Ship and Island platforms during ORCESTRA, collected to Decipher the ITCZ" by Winkler et al. (2026).*

*This dataset, conducted by a collaborative team over ocean and island platforms, presents a potentially valuable resource for investigating fine-scale vertical atmospheric structures. The experiment itself appears both well-designed and scientifically engaging. Compared to other ocean-based sounding campaigns, the launch frequency in this study is notably higher in temporal density, which should offer more detailed perspectives for studying the Intertropical Convergence Zone (ITCZ). Overall, the paper is well-organized and provides thorough technical documentation. This research holds great potential, and the dataset itself possesses significant scientific value. I have benefited from the authors' academic rigor and would like to express my appreciation for their thoughtful work. I would recommend major revision to the manuscript prior to publication.*

We thank the Reviewer for the feedback and address the individual points below.

*1. As a data description paper, I found it quite difficult to access the data link provided in the Abstract. Many readers may not be familiar with the IPFS system. Providing an alternative HTTP address would be more appropriate. Since the author provides corresponding Python code later in the text, it might be better to include a brief explanation in the abstract.*
* * *
[1] https://doi.org/10.5194/essd-2025-638-RC1

IPFS was deliberately chosen for this campaign as it provides a decentralized, content-addressable data distribution mechanism that is well suited for large, multi-platform field campaigns coordinated by multiple institutions across different countries. By referencing datasets via a unique content identifier (CID), IPFS ensures data integrity and long-term reproducibility, while avoiding dependence on a single institutional server and reducing the risk of broken or unavailable links.

In previous campaigns of comparable scope, data were often distributed across institution-specific servers or personal storage systems, resulting in fragmented access and limited long-term availability. The use of IPFS in ORCESTRA enabled the community to collectively share a unified dataset in a robust and platform-independent manner from an early stage of the project.

Nevertheless we agree that many readers may be unfamiliar with IPFS. To improve accessibility, we added a more precise explanation in the code availability section in addition to the Python example code provided later in the text.

*2. To my knowledge, there have been many other Atlantic radiosonde launch campaigns, such as POLARSTERN, DBLK, HTXUH4H, among others. On an old hard drive, I discovered approximately 20,000 high-resolution radiosonde profiles launched over the ocean, most of which were conducted over the Atlantic. As an example, I have provided two screenshots of the data listing below. Although I have not systematically examined the spatial overlap between these data and the study area, it might be worthwhile to briefly introduce other oceanic radiosonde experiments in the introduction and provide a concise comparison. This would allow readers to gain a broader perspective on the full scope of Atlantic radiosonde campaigns.*

Figure 1: Screenshot 1

[Figure]

Figure 2: Screenshot 2

We acknowledge the long-standing and important contributions of ship-based radiosonde launches, including routine and campaign-based observations from research vessels, which have been crucial for advancing the understanding of the tropical atmosphere over the Atlantic.

However, ORCESTRA was designed with a different observational strategy. Rather than focusing on a single platform or cruise, the campaign implemented a coordinated, multi-platform network including island stations, a research vessel, and aircrafts operating simultaneously across the tropical Atlantic over an extended period. The spatial coordination and temporal sampling strategy are therefore not directly comparable to single-platform cruises or long-term routine launches. For this reason, in the introduction we compare ORCESTRA only to similarly coordinated, multi-platform field campaigns.

A comprehensive overview paper of the ORCESTRA campaign is currently in preparation and will place ORCESTRA in the broader historical context of Atlantic radiosonde observations, including previous ship-based efforts. To avoid unnecessary repetition, this perspective is not expanded further in the present data description paper.

*3. The authors emphasize the ITCZ in their title, yet more detailed analysis of the ITCZ is not found in the main text. If feasible, providing a preliminary finding on the ITCZ could enhance the scientific contribution of this paper.*

While a comprehensive analysis of the ITCZ is not the primary aim of this data paper, we amended the summary section to explicitly elaborate on the scientific possibilities enabled by the dataset with respect to ITCZ research.

In particular, the RAPSODI radiosonde observations provide high-vertical-resolution profiles of temperature, humidity, and winds across repeated meridional transects and at both the center and margins of the Atlantic ITCZ. These measurements enable analyses of moisture convergence, vertical wind shear, and the thermodynamic structure relevant to ITCZ position, width, and variability. When combined with complementary ORCESTRA observations the dataset forms a basis for constraining circulation regimes within and adjacent to the ITCZ.

The hypothesis-driven analysis of ITCZ dynamics will be presented in forthcoming dedicated campaign papers, where the radiosonde dataset will be integrated with aircraft, ship-based, and remote-sensing observations.

*4. The abstract should include more details about the radiosonde dataset, such as the time range, release intervals, and balloon sampling frequency. Additionally, the R/V Meteor may be difficult to understand for readers unfamiliar with German scientific expeditions; for instance, I initially assumed it referred to a type of meteor radar.*

We have added further details on the radiosonde dataset, including the campaign time period and typical launch frequency to the abstract. Regarding the R/V Meteor, this is the official name of the German research vessel and is widely used in the literature. To avoid potential confusion for readers unfamiliar with German research expeditions, we now explicitly introduce it as the German research vessel R/V Meteor at its first mention in the abstract and main text.

*5. The Introduction would benefit from a clearer explanation of the scientific motivation behind ORCESTRA. What were the key research questions or atmospheric processes that this campaign aimed to address?*

The scientific motivation and key research questions of ORCESTRA are described in detail in the dedicated ORCESTRA overview paper, which is cited in the first paragraph of the introduction (Stevens et al., Tellus, 2026, submitted). In addition, each subcampaign contributing to ORCESTRA (PERCUSION, MAESTRO, BOWTIE, PICCOLO, and SCORE) addresses specific scientific questions and will document its individual motivation and objectives in separate publications. To keep the present manuscript focused on the description of the radiosonde dataset, we therefore limit the discussion of scientific motivation here and refer readers to the overview and subcampaign papers for a comprehensive treatment.

*6. It is recommended that the following content from page 2 be annotated within the main text. This citation format appears inconsistent with standard EGU citation style and caused some confusion. Additionally, page 3 contains similar phrasing such as "MAESTRO (mesoscale organization of tropical convection)," which duplicates the expression on page two.*
* * *
[1]https://orcestra-campaign.org/orcestra.html
[2]PERCUSION ≡ Persistent EarthCare underflight studies of the ITCZ and organized convection
[3]MAESTRO ≡ Mesoscale organisation of tropical convection
[4]BOWTIE ≡ Beobachtung von Ozean und Wolken – Das Trans ITCZ Experiment
[5]PICCOLO ≡ Process Investigation of Clouds and Convective Organization over the atLantic Ocean
[6]SCORE ≡ Sub-Cloud Observations of Rain Evaporation

We revised the introduction by removing footnotes and avoiding inline expansions of multiple acronyms. The ORCESTRA subcampaigns are now summarized in table 1 that lists all acronyms and their full names.

*7. Figure 1a: Why do some trajectory segments appear discontinuous in the lower-right part of the panel? This is unusual in my experience—could it be due to data loss?*

The apparent discontinuities previously visible in the lower-right part of panel (a) were associated with a small number of radiosonde trajectories launched from R/V Meteor toward the end of its Atlantic transect. For clarity they have now been removed from inset (a) in Figure 1.

Each trajectory represents a single radiosonde flight, with points showing the sonde's horizontal position at different altitudes and colors indicating height, such that ascent and post-burst descent appear as a continuous blue–green–blue progression. From a top-view perspective, trajectories may appear irregular or segmented because horizontal drift is governed by

altitude-dependent winds, which can produce curved or meandering paths when shown as discrete position markers. Occasional brief data gaps due to telemetry loss are unavoidable in radiosonde observations and are represented in the dataset as missing (NaN) values.

*8. For panels (a) and (b), which represent land-based ("stationary") platforms, adding launch coordinates to the figure would be helpful.*

We revised Figure 1 and have added launch location markers and coordinate annotations for the land-based platforms shown in panels (a) and (b). The nominal launch locations are indicated by red crosses, while two launches from slightly displaced locations at INMG are shown with black crosses.

*9. Additionally, I made every effort to interpret Figure 1, as it is crucial for understanding the entire experiment. Unfortunately, despite over a decade of radiosonde experience, I find Figure 1 difficult to comprehend. The phrase "...both ascending and descending segments shown" is perplexing. Typically, descending balloons, as described in the main text, are called dropsondes, while ascending ones are radiosondes. I'm unclear on what exactly the "descending" values represent. Does "descending" refer to the period after balloon burst? A typical radiosonde profile (sampling rate 1s) would produce a continuous curve rather than the scattered points shown in (b).*

In this study, the terms ascending and descending are used to distinguish the two branches of a single balloon-borne radiosonde flight. The ascending segment refers to measurements collected while the radiosonde is carried upward by the balloon, whereas the descending segment refers to measurements collected after balloon burst, when the same instrument descends under a parachute.

Within ORCESTRA, we reserve the term dropsonde for instruments released from aircraft that measure only a downward trajectory. In contrast, the radiosondes discussed here are balloon-launched and may provide both ascending and descending profiles from the same sensor. The "descending" values shown in Figure 1 therefore correspond to post-burst measurements, not to aircraft-deployed dropsondes. We have clarified the terminology in the manuscript.

While radiosondes typically sample at 1 Hz, Figure 1 does not display every sampled point. Instead, a subset of positions is plotted to avoid overloading the figure and to better convey the spatial distribution and density of coverage across the eastern, central, and western Atlantic. The intent of this visualization is to illustrate the geographic extent and overlap of radiosonde trajectories rather than to represent individual profiles at full temporal resolution.

*10. Given that Meteomodem and Vaisala radiosondes are well-established and widely documented technologies (e.g., in journals like AMT), the authors might consider reducing the technical details in favor of highlighting the unique scientific opportunities offered by the ORCESTRA campaign. What are the potential research themes enabled by this dataset? Which atmospheric processes could be better examined? Expanding the Summary to include such perspectives would increase the impact and value of the data.*

Due to the multi-vendor and multi-platform nature of the ORCESTRA campaign, we consider it valuable to document the technical implementation, processing choices, and quality-control steps of the radiosonde observations in a single, consolidated data paper. To better highlight the scientific value of the observations, we have expanded the summary section to motivate possible research themes and questions that can be addressed using the radiosonde dataset.

**RAPSODI: Radiosonde Atmospheric Profiles from Ship and island platforms during ORCESTRA, collected to Decipher the ITCZ**

Marius Winkler, Marius Rixen, Florent Beucher, Fleur Couvreux, Chaehyeon C. Nam, Philippe Peyrillé, Hauke Schmidt, Hans Segura, Karl-Hermann Wieners, Ezri Alkilani-Brown, Abdou Aziz Coly, Giovanni Biagioli, Michael M. Bell, Ester Brito, Emma Chauvin, Julie Capo, Delián Colón-Burgos, Akeem Dawes, Jose Carlos da Luz, Zekican Demiralay, Vincent Douet, Vincent Ducastin, Clarisse Dufaux, Jean-Louis Dufresne, Florence Favot, Thomas Fiolleau, Emilie Fons, Geet George, Helene M. Gloeckner, Suelly Gonçalves, Laurent Gouttesoulard, Lennéa Hayo, Wei-Ting Hsiao, Sarah Kennison, Michael Kopelman, Tsung-Yung Lee, Enora Le Gall, Mateo Lovato, Emily Luschen, Nicolas Maury, Brett McKim, Louis Netz, Diouf Ousseynou, Karsten Peters-von Gehlen, Chavez Pope, Basile Poujol, Niwde Rivera Maldonado, Nina Robbins Blanch, Nicolas Rochetin, Daniel Rowe, Paula Romero Jure, James H. Ruppert Jr., Jairo Segura Bermudez, Jarrett C. Starr, Martin Stelzner, Connor Stoll, Macintyre Syrett, Abraham Tekoe, Jeremie Trules, Colin Welty, Daniel Klocke, Raphaela Vogel, Sandrine Bony, Allison A. Wing, Bjorn Stevens

February 7, 2026

**Response to RC-2[1]**

*Review of "RAPSODI: Radiosonde Atmospheric Profiles from Ship and Island platforms during ORCESTRA, collected to Decipher the ITCZ" by Winkler et al. (2026).*

*The data paper by Winkler et al. describes the radiosonde data set collected during the ORCESTRA field campaign, consisting of three different sounding operations.*

*The paper gives a good overview of the data collect and the processing steps between the raw data and the final data produces. The manuscript describes where the data can be accessed and their formats. The Level 0 data can be accessed directly, the higher level zarr data require more specialized knowledge.*

*The processing steps are clear and appropriate, and I expect that data quality is of good quality.*

*I recommend publication of this manuscript after minor revisions.*

We thank the Reviewer for the feedback and address the individual points below.

*1. The processing uses the vapor pressure equation of IAPWS (Wagner and Pruss, 2002). For radiosonde data, the vapor pressure equation used by the manufacturer should also be used in subsequent calculations to avoid any discrepancies between the calibration of the sensor and the measurements. At warm temperatures, this issue is irrelevant; however, at cold temperatures near the tropopause, this issue may become quite significant.*
* * *
[1] https://doi.org/10.5194/essd-2025-638-RC2

*Vaisala uses the equation by Hardy (1998) and Modem uses the equation by Sonntag (1994). Fortunately, the differences to the IAPWS formulation are small and reprocessing is not needed, but you should convince yourself of this difference and make a statement to that effect.*

We compared several commonly used formulations of the saturation vapor pressure over liquid water, including Wagner and Pruss (2002), Hardy (1998), Sonntag (1990, 1994), Murphy and Koop (2005), and others, against the IAPWS-97 reference formulation (c.f. Figure 1). The formulations by Wagner–Pruss, Hardy, Sonntag, and Murphy–Koop all exhibit very close agreement with the IAPWS-97 reference, with deviations typically well below $1\%$. In contrast, more commonly used Magnus- or Tetens-type approximations (e.g. Bolton, 1980) show substantially larger deviations.

Below the formal IAPWS-97 validity range, no definitive reference exists. The clustering of deviations among simpler empirical formulations at low temperatures therefore does not imply a systematic bias in Wagner–Pruss, particularly given their inferior performance at higher temperatures and their limited physical basis. The closer agreement of Murphy–Koop with these formulations at low temperatures instead reflects its specific optimization for ice saturation.

We conclude that replacing the manufacturer-specific formulations (Hardy for Vaisala and Sonntag for Meteomodem) with a single, consistent formulation does not introduce a meaningful bias in the derived humidity-related quantities for this dataset.

[Figure]

Figure 1: Relative differences of commonly used saturation vapor pressure formulations over liquid water with respect to the IAPWS-97 reference, shown as $|e_{s,x}/e_{s,ref} - 1|$ on a logarithmic scale.

*2. Likewise, the inverse vapor pressure equation (1) to calculate the dew point temperature is a reasonable inversion of the Hardy equation used by Vaisala. However, it is NOT a good inversion of any Magnus-type relation. You should clarify that this is a good approximation of the inversion of the equation by Hardy (1998). Can you also provide a reference for this equation.*

We agree that the inverse vapor pressure relation used here is not a general inversion of

a Magnus-type formulation. The dew-point equation is an empirical approximation designed to invert the saturation vapor pressure formulation by Hardy (1998), which is used in the Vaisala MW41 processing chain. We have clarified this explicitly in the manuscript and added the appropriate reference.

*3. Table 1 should include the exact radiosonde model and its weight, especially for the Modem sonde. Also include the software version for the MW41 and Eoscan software.*

    Table 1 has been updated.

*4. Figure 2 is oriented like a Hovmöller diagram. It would be easier to read if the stations were along the vertical axis and time increasing along the horizontal axis. It will use less space in the final paper.*

    The figure is intentionally oriented this way so that it fits within a single column in the final two-column layout, ensuring good readability without requiring a full-page figure. For this reason, we prefer to retain the current orientation.

*5. Line 101: Why were the additional radiosonde handled differently, i.e., placed outside before being attached to the balloon inside? I assume the same launcher was used for all sondes.*

    The difference in handling was not related to the launcher, which was the same for all radiosondes, but solely to the configuration of the two ground receivers used during the campaign. For both receivers deployed on the R/V Meteor, the radiosondes had to be brought briefly outside the container to establish a satellite telemetry connection due to lack of reception inside the respective container. However the initialization process prior to this was different for the two receivers. For the MPI receiver, the radiosonde was first initialized indoors using a tray-like adapter connected to the receiver, after which it was taken outside to acquire the satellite signal. The DWD receiver did not use such an adapter. In this case, the radiosonde was switched on manually using the button on the sonde itself and placed outside to complete its initialization and establish the satellite connection. After successful initialization, the radiosondes were brought back inside for balloon attachment and launch. We have reformulated the manuscript accordingly for clarity.

*6. Line 105: I assume you mean "telemetry link" instead of "connection".*

    Yes, that is more precise and therefore better.

*7. Line 113: Instead of "… drifted farther east of the Atlantic than in the west" write "drifted farther in the eastern than in the western the Atlantic"*

    Incorporated.

*8. Figure 4: Please make the left-hand Figure wider so that the biases can be read better. You can achieve this by moving the legends.*

    Incorporated.

*9. Line 154: Instead of "Apart from six post-launch contact losses" better write "Apart from telemetry loss of six sondes"*

    Incorporated.

*10. Line 166: Only icing is a risk prevented by the heater. Saturation is an atmospheric state.*

    We incorporated your suggestion.

*11. Line 173: This is a repeat from line 165.*

    We shortened the sentence to omit repetition.

*12. Line 192: replace "deployed" with "opened"*

Incorporated.

*13. Figure 7: The 90th percentile of the descent rate is close to zero for both Modem and Vaisala sondes. In additional there seems to be a kink in the mean descent rate for the Modem sondes at around 20 km. This is extremely unlikely a real behavior and more likely either a processing issue or potentially a GPS receiver issue. You should look into the fall rate of the sondes and understand what causes this odd distribution.*

[Figure]

Figure 2: Vertical profiles of temperature at INMG (red) and BCO (blue). The tropopause is more sharply defined at INMG, as indicated by the stronger vertical temperature gradient compared to BCO.

The near-zero values of the upper percentiles reflect the wide spread of descent speeds, where a subset of soundings exhibits very slow vertical motion. The apparent kink in the mean descent rate of the Meteomodem profiles around 20 km coincides with the transition across the tropical tropopause, where changes in static stability, air density, and oscillatory motion modify the descent behavior. Such behavior has been discussed in the context of ascent–descent differences and radiosonde behavior near the tropopause (e.g. Dupont et al., 2020). Another detailed analysis of RS41 descent data by Ingleby et al. (2022) reports a comparable inflection between roughly 15 and 20 km and attributes it to descent dynamics rather than processing or GPS artifacts. Although that study focuses on Vaisala RS41 sondes, which descend under a parachute, the underlying transition is not instrument-specific. In contrast, Meteomodem sondes in this campaign descended without a parachute and may therefore be more directly influenced by this dynamical change. The slightly more pronounced inflection observed for Meteomodem descent rates may thus indicate an enhanced sensitivity to the changing flow regime rather than a sonde-specific processing issue. As shown in Fig. 2, the tropopause is more sharply defined at INMG than at BCO, as indicated by the stronger vertical temperature gradient, which likely enhances the response of descending sondes when crossing this layer. Independent analyses from other tropical sites employing Meteomodem radiosondes without parachutes (e.g.

Faa'a, French Polynesia and Nouméa, New Caledonia) show a similar inflection near 20 km, further supporting a physical rather than instrumental origin. We have clarified this also in the manuscript.

*14. Line 195 and supplemental material. You discuss the supplemental Figures in the main text. Since there are only three of them and they contribute to the overall paper, I would suggest you include all in the main manuscript. In Figure S1 and Figure 8, panel c each, I would suggest changing the units to [%], which is more common and easier to read.*

Incorporated.

*15. You use the RH units of % in equation (1), but not in equation (2).*

We amended equation (2).

*16. Line 198: Change "profile" to "sounding".*

Incorporated.

*17. Line 199: Is the splitting into ascent and descent profile on geopotential altitude, or GPS altitude grid? There is a space missing after 10 m.*

Ascent and descent profiles are separated using the radiosonde dropping flag, not by altitude. The profiles are subsequently binned on a 10 m geopotential-height grid.

*18. Line 225: This is confusing. Does the M20 sonde also carry a pressure sensor?*

The M20 radiosonde includes a barometric pressure sensor, which is used to detect the start of the sounding and during the initial ascent (approximately the first 1000 m). For the remainder of the profile, pressure is derived from GPS altitude rather than from the onboard barometer. The text has been clarified accordingly.

*19. Table 2: What is meant by "minimal vendor processing"? I assume that the full vendor processing including smoothing, radiation correction, and time response correction for humidity has already been applied at that stage.*

The term "minimal vendor processing" was imprecise. At that stage, the data include the full standard vendor processing. The wording has been revised accordingly.

*20. Table 2 and Line 243: In the normal Level 1 processing of Vaisala radiosondes, which XML tables did you use?*

For Vaisala radiosondes, Level-1 quantities are read from the XML tables Synchronized-SoundingData.xml, Soundings.xml, and Radiosondes.xml. The manuscript has been amended to clarify this.

*21. Line 229: Both Vaisala and Modem generate more file than just .cor or .mwx. These systems can generate additional ASCII text files as well as bufr or temp files, which are distributed to the GTS. Can you describe all files and whether bufr or temp files were distributed to the GTS. That information may be useful if any user would like to use these data to compare against reanalyses, which may use these data.*

For this campaign, the data handling strategy was intentionally focused on using .cor files for Meteomodem and .mwx files for Vaisala. These formats contain the full set of measurement, metadata, and processing information required for subsequent harmonization and conversion into the distributed NetCDF products.

Regarding GTS dissemination, the handling differed by platform and receiver configuration. On board the R/V Meteor, soundings launched at 00, 06, 12, and 18 UTC using the permanent DWD receiver were transmitted operationally to the GTS as SYNOP and TEMP messages under

the ship identifiers DBBH (until 9 September 2024) and GNJ7EFP (from 10 September 2024), with TEMP data additionally disseminated under the identifier ZVQEQCM. These data were received and processed by ECMWF and may therefore have been assimilated into operational analyses and reanalyses. In contrast, soundings launched at 03, 09, 15, and 21 UTC using the MPI receiver were not transmitted to the GTS, as automatic GTS upload was not enabled for this system, neither on the R/V Meteor nor at the BCO.

Radiosonde data from INMG on Sal were not distributed to the GTS in real time. After the campaign and following quality control, the Meteomodem soundings launched from INMG were converted to BUFR format and transmitted in delayed mode to ECMWF. These data may therefore be assimilated in future reanalysis products.

We have clarified this explicitly in the manuscript.

*22. Line 265: Please provide the values for Tc and Pc.*

The values have been added to the manuscript: the critical temperature and pressure of water vapor are $T_c = 647.096\,\mathrm{K}$ and $P_c = 22.064 \times 10^6\,\mathrm{Pa}$, respectively.

*23. Line 271: Masking relative humidities > 100% prior to any binning or averaging can lead to small negative biases.*

The masking step was included as a precautionary consistency check. However, in the dataset used for this study no relative humidity values exceeding 100% were found. Consequently, this step does not affect the binning or averaging.

*24. Line 305: Please give the values for Rd and Cpd.*

The manuscript has been updated to specify the values used: $R_d = 287.05\ \mathrm{J\,kg^{-1}\,K^{-1}}$ and $c_{p_d} = 1004.0\ \mathrm{J\,kg^{-1}\,K^{-1}}$.

*25. Line 315: Why do you exclude the soundings while the R/V Meteor is still in Mindelo? After the arrival at Barbados, you showed that there is very little difference between the ship borne and land-based soundings. In addition, it would be interesting to see a comparison between the Modem and Vaisala soundings at Cabo Verde similar as that you did in Barbados.*

The soundings launched while the R/V Meteor was still in Mindelo are excluded only from the IWV analysis in order to focus on open-ocean conditions. These launches occurred while the vessel was in port, were conducted sporadically rather than following the standard 8-per-day schedule, and were not accompanied by other shipborne measurements. Including them would introduce a small number of profiles influenced by coastal conditions, leading to a slightly drier tail in the IWV distribution that is not representative of the open Atlantic. The comparison between shipborne and land-based soundings at Barbados addressed a different question and was intended to assess platform effects rather than IWV characteristics. The R/V Meteor was stationed in coastal waters approximately 1.5 km offshore from the Barbados Cloud Observatory, with the regular 8-per-day launch schedule and concurrent measurements maintained, enabling a meaningful comparison. A similar comparison in the eastern Atlantic was not pursued, as the vessel did not approach the INMG site with comparable spatial proximity.

*26. Line 326: For all sounding sites, how is the first line of the radiosonde profile handled? Is it a measurement from a nearby surface reference station, or the potentially biased radiosonde data, which is not yet properly ventilated? If an external surface station is used, then all profiles should have a surface value.*

The first level of each raw (Level-0) radiosonde profile corresponds to surface observations taken immediately prior to launch. For land-based sites, this first line is provided by collocated

surface reference stations (e.g. BCO and the INMG station[2] for MAESTRO), while for ship-based launches it corresponds to the onboard meteorological instrumentation. These surface values are therefore present in all profiles. An exception applies to two MAESTRO soundings launched from a slightly offset location (black crosses in Figure 1 inset b)), for which the first level corresponds to the radiosonde measurements at launch and may be subject to incomplete ventilation.

The Level-2 product used in the IWV computation here in line 326, is obtained by vertically resampling the raw profiles onto a fixed altitude grid (0–31 km). Because the actual launch altitude differs between platforms and sites (e.g. land stations versus ship deck), the lowest valid measurement in the resampled Level-2 profile is not necessarily located at 0 m. As a result, the surface reference measurement present in the raw profile may appear at a higher altitude level in the Level-2 product, or may not coincide with the lowest grid level. Consequently, not all Level-2 profiles contain a valid value at 0 m. If near-surface humidity values are missing at the lowest grid levels, limited backfilling is applied to enable the computation of IWV.

*27. Similarly, a temporary setup of a system is often prone to small configuration errors. Did you compare the settings of the second sounding system on the ship with the permanently installed system? Are all heights correct, and are the launches detected properly without loss of data just after launch? You can verify this by comparing the raw and processed data in the mwx file.*

On board R/V Meteor, the configuration of the temporarily installed sounding system was cross-checked against the permanently installed system. Height measurement used for radiosonde initialization were taken from the same shipboard instrument in both systems. Data integrity immediately after launch was routinely monitored, with each launch observed for at least 10 min to ensure stable telemetry, correct launch detection, and continuous data reception.

*28. Lines 357: Descending profiles are often not tracked to the surface, which makes it more difficult to get a good integrated water column. You could create Figure 10 using only ascent profiles. This would also reduce any hysteresis or other sensor issues that may exist on descent.*
* * *
[2]https://oscar.wmo.int/surface/index.html#/search/station/stationReportDetails/0-20000-0-08594

[Figure]

Figure 3: Comparison of IWV distributions using ascent-only profiles and combined ascent–descent profiles.

We agree that descending profiles are more prone to incomplete near-surface coverage and potential sensor hysteresis effects. For this reason, we applied the strict IWV filtering criteria described in Section 3.3 to ensure physically meaningful column integrals for both ascent and descent segments. However, we repeated the analysis using only ascent profiles. As shown by the comparison in Figure 3 in this document, the resulting IWV distributions remain qualitatively unchanged, with only marginal quantitative differences. The central tendencies and relative offsets between platforms are preserved. We therefore retain the combined ascent–descent dataset in order to maximize the use of available observations, while noting that the applied filtering effectively mitigates the issues raised for descent profiles.

*29. Line 362: "... reaching the lowest ..."*

Incorporated.

*30. Data availability: The Level 0 data can be accessed as described. The information provided*

*is not sufficient to access the .zarr data. You should provide more information regarding the IPFS since access without does not seem to be possible. Some implicit python libraries are missing and better directions for installing an IFPS client would be useful. Access through the web page is possible.*

We have revised and expanded the Data and Code Availability section to clarify access to the `zarr` datasets. We now provide step-by-step guidance for programmatic access via IPFS, list the required Python dependencies, and include an example installation command. We hope that these additions make data access straightforward for users with varying levels of technical experience.